# Investigation of CD56, ADAM17 and FGF21 Expressions in the Placentas of Preeclampsia Cases

**DOI:** 10.3390/medicina59061145

**Published:** 2023-06-14

**Authors:** Irem Darka Aslan, Gorker Sel, Figen Barut, Rabia Baser Acikgoz, Sibel Balci, Ulku Ozmen, Aykut Barut, Muge Harma, Mehmet Ibrahim Harma

**Affiliations:** 1Department of Gynecology and Obsterics, Faculty of Medicine, Zonguldak Bulent Ecevit University, Esenköy, Kozlu, 67000 Zonguldak, Turkey; iremdarka.id@gmail.com (I.D.A.); gorkersel@gmail.com (G.S.); ulkuozmen1@gmail.com (U.O.); aykbar@yahoo.com (A.B.); obgyn.info@gmail.com (M.H.); gtd.contact@gmail.com (M.I.H.); 2Department of Medical Pathology, Faculty of Medicine, Zonguldak Bulent Ecevit University, Esenköy, Kozlu, 67000 Zonguldak, Turkey; 3Department of Gynecology and Obsterics, Zonguldak Alapli Public Hospital, Yeni Siteler Street, 67850 Zonguldak, Turkey; 4Department of Biostatistics, Faculty of Medicine, Kocaeli University, 41380 Kocaeli, Turkey; b.s.balci@gmail.com

**Keywords:** ADAM17, CD56, FGF21, placenta, preeclampsia

## Abstract

*Objective:* In the present study, we investigated the expression of CD56, ADAM17 and FGF21 antibodies (Ab), which we think have an effect on the pathophysiology of preeclampsia (PE), in pregnant patients with healthy placentas and placentas with PE. The expression of these antibodies has been investigated in a limited amount of former research, but their role in PE has not yet been clarified. With this study, we aimed to contribute to the elucidation of the pathophysiology of PE and the detection of new target molecules for treatment. *Materials and Methods:* Parturients with singleton pregnancy at 32 weeks or above without any maternal or fetal pathology who were admitted to the Department of Obstetrics and Gynecology, Zonguldak Bülent Ecevit University Practice and Research Hospital between 11 January 2020 and 7 January 2022 were included in the present study. Pregnant women with coexisting disease or a pathology related to the placenta (ablation placenta, vasa previa, hemangioma, etc.) were excluded. CD56, ADAM17 and FGF21 antibodies were histopathologically and immunohistochemically detected in 60 placentas with PE (study group) and 43 healthy placentas (control group). *Results:* CD56, ADAM17 and FGF21 proteins were all more intensely expressed in preeclamptic placentas and a statistically significant difference was found between the two groups for all three antibodies (*p* < 0.001). Deciduitis, perivillous fibrin deposition, intervillous fibrin, intervillous hemorrhage, infarct, calcification, laminar necrosis and syncytial node were found to be significantly more common in the study group (*p* < 0.001). *Conclusions:* We observed that CD56, ADAM17 and FGF21 expressions increased in preeclamptic placentas. These Ab may be responsible for the pathogenesis of PE, which can be illuminated with further studies.

## 1. Introduction

Preeclampsia (PE) is a multisystemic progressive disease characterized by new-onset hypertension in the last half of pregnancy or after delivery, and end-organ dysfunction with or without proteinuria [1]. It is a clinical condition where systolic blood pressure (SBP) is >140 mmHg and diastolic blood pressure (DBP) is >90 mmHg in at least two consecutive measurements with 4 h intervals after the 20th gestational week, and is accompanied by end-organ hazard. If the measured SBP values are above 160 mmHg, DBP above 110 mmHg and at least one organ damage is accompanied, it is called severe PE [2]. Although its pathogenesis has not been clearly elucidated, many pathophysiological mechanisms thought to cause PE have been suggested. Conditions that cause abnormal remodeling of spiral arteries, defective trophoblast differentiation, oxidative stress, placental hypoperfusion and ischemia cause abnormal placentation. In addition, immunological factors, genetic factors, environmental factors, inflammation, increased angiotensin-II sensitivity and endothelial dysfunction are also involved in the pathogenesis of PE [3,4].

There are limited studies investigating the relationship of Cluster of differentiation 56 (CD56), A disintegrin and metalloprotease 17 (ADAM17) and Fibroblast growth factor 21 (FGF21) with hypertension and PE. In our study, we studied the expressions of CD56, ADAM17 and FGF21 antibodies, which we think will be useful in illuminating the pathophysiology of PE, in placenta samples of both healthy and PE-diagnosed pregnant women.

CD56, also known as a neural cell adhesion molecule, is the junction glycoprotein of natural killer (NK) cells. CD56 expression is present in intravascular trophoblasts and decidual NK cells. Trophoblast invasion and the remodeling of spiral arteries are regulated by uterine NK cells. It has been discovered that trophoblastic remodeling at early implantation occurs via the differentiation of trophoblasts to become CD56 immunoreactive cells. NK cells are believed to promote placental and trophoblastic growth and provide immune modulation at the maternal–fetal interface during pregnancy. CD56 immunostaining is used as an additional marker to identify intravascular trophoblasts and decidual vasculopathy. Various studies in the literature have shown that acute atherosis, fibrinoid medial necrosis and decidual vasculopathy can be identified usingCD56 immunostaining [5,6,7].

ADAM17 is a protease responsible for releasing the extracellular domains of transmembrane proteins, thereby producing paracrine and autocrine signaling molecules [8]. Ischemia and oxidative stress resulting from insufficient trophoblast invasion cause an increase in ADAM17, and an increase in ADAM17 induces an increase in Tumor necrosis factor-alpha (TNF-α). Increased TNF-α in response to local hypoxia plays an important role in placental oxidative stress and PE-related endothelial damage.

ADAM17 substrates include adhesion proteins, membrane-bound cytokines (TGF-α) and cytokine receptors (TNF-α receptor, IL-6 receptor) [9]. ADAM17 plays a role in proinflammatory and profibrotic pathways, diabetes mellitus (DM), cardiovascular diseases, chronic kidney disease, inflammatory diseases and other pathological processes [10].

The most active metabolite of the Renin-Angiotensin System (RAS) is angiotensin II (Ang-II), which promotes vascular injury and hypertension primarily through interaction with the Ang-II type 1 receptor (AT1R). There is evidence of increased activation of AT1R during RAS hyperactivity via Ang-II, which promotes translocation of ADAM17 to the cell membrane [11]. Changes induced by ADAM17-induced cytokines such as angiotensin-converting enzyme-2 (ACE-2) and other membrane proteins can lead to endothelial dysfunction, which in turn alters the vascular environment by inducing vascular hemodynamics, endothelial damage, barrier dysfunction and dysregulation of vascular tone control. Our study will help to evaluate its relationship with the increased expression of ADAM17, which provides the release of cytokines, especially TNF-α, due to oxidative stress in the preeclamptic placenta.

FGF21 is a hormone secreted mainly from the liver, pancreas, brown and white adipose tissue and also from other metabolically active tissues such as the placenta during pregnancy. FGF21 stimulates glucose uptake in adipose tissue. However, it does not stimulate other cell types. This effect contributes to insulin activity. It was reported that FGF21 has many beneficial effects on main cardiovascular risk factors such as hyperlipidemia, obesity and DM due to its ability to regulate carbohydrate and lipid metabolism [12]. However, contrary to the observations in animal models, there are also studies that reported high circulating FGF21 levels in humans to be associated with metabolic syndrome and dyslipidemia [13].

In recent studies, it has been revealed that FGF21 levels increase in the blood of pregnant women with PE before 28 weeks [14,15,16]. However, the expression of placental FGF21 in PE and whether it has an effect on altered placental metabolism is not reported in the literature.

Since elucidating the pathophysiology of PE will also contribute to treatment strategies, we aimed to examine the immunohistochemical expressions of these molecules, which we think have an effect on the pathophysiology, in healthy pregnant placentas and pregnant placentas with PE.

## 2. Materials and Methods

The present study included 103 pregnant patients. The study was carried out in accordance with the Declaration of Helsinki and was approved by the Ethics Committee of the Bülent Ecevit University Faculty of Medicine (2020/18-14, 16 September 2020). Informed consent was obtained from all pregnant women included in the study. The study group included 60 patients with a single pregnancy of 32 weeks and above with a diagnosis of PE who were admitted to the Department of Obstetrics and Gynecology, Zonguldak Bülent Ecevit University Practice and Research Hospital between November 2020 and July 2022. The control group included 43 patients with singleton pregnancy at 32 weeks or above without any maternal or fetal pathology. Patients with an additional disease such as chronic hypertension, DM, gestational DM, hypothyroidism, asthma/bronchitis or a pathology related to the placenta (ablation placenta, vasa previa, hemangioma, etc.), pregnant women who are smokers, who are obese and IVF pregnancies were not included in the study. Parturients with a history of PE in their previous pregnancy were also excluded to clarify the role of CD56, ADAM17 and FGF21 Ab in the pathogenesis of new onset PE. Placentas were sampled for histopathological and immunohistochemical examination.

### 2.1. Histopathological Examination

After the placental tissues in the study and control groups were fixed with 10% neutral buffered formaldehyde, appropriate sampling was made from the placenta of each case and paraffin blocks were prepared after routine follow-up. Hematoxylin-and-eosin (H&E)-stained preparations were obtained from paraffin blocks. These preparations were evaluated by a single pathologist (FB) with a light microscope (Leica DM2500 microscope, Leica Microsystems, Wetzlar, Germany) by semiquantitatively scoring the histopathological findings showing maternal hypoxia, blinded for the study groups. Histopathological findings showing maternal hypoxia, Perivillous and intervillous fibrin deposition, intervillous hemorrhage, placental infarcts, placental calcification, syncytial nodes, distal villous hypoplasia, placental regional giant cell increase, laminar necrosis and chorionic pseudocysts were examined under light microscope and scored as; 0: no histological change, 1: mild histological change, 2: significant histological change, 3: severe histological change [17,18].

### 2.2. Immunohistochemical Method

Immunohistochemical examination was performed on 4–6 μm thick sections prepared from paraffin blocks of tissues fixed in 10% formalin solution. Immunohistochemical application for each Ab was applied to all cases in a single session and positive tissue control was used.

Paraffin sections of 4–6 μm thickness, prepared on an adhesive slide, were kept in an oven heated to 80 °C for one hour, and immunohistochemical studies were performed with a Ventana Benchmark Ultra automatic staining device (Ventana Medical Systems Inc., Tuscon, AZ, USA) in accordance with staining protocols.

CD56 (Mouse monoclonal, Clone 123C3.D5, 1/150, Cell Marque, Rocklin, CA, USA), Anti-ADAM17 (Rabbit polyclonal, ab39162, 1/150, abcam, Cambridge, UK) and Anti-FGF21 (Rabbit monoclonal, Clone EPR8314(2), ab1711941, 1/250, abcam, Cambridge, UK) primary antibodies and the staining intensities of these primary antibodies were used, determined by an independent single pathologist (FB) and scored using semi-quantitative evaluation under the light microscope.

Semiquantitative scoring of the immune reaction was performed according to the following criteria:

If the staining intensity was less than 10%: 0 (zero);

Between 10 and 25%: 1 (mild);

Between 25 and 50%: 2 (medium);

More than 50%: 3 (severe).

Since the patients with less than 10% staining dominated the other three groups, statistical analysis could not be performed. Therefore, the groups were classified as pairs; the groups stained at 0–1 intensity were classified as low expression, and the groups stained with 2–3 intensity were classified as high expression groups.

### 2.3. Statistical Analysis

Statistical analysis was performed with the IBM SPSS 20.0 (IBM Corp., Armonk, NY, USA) program. Normal distribution was evaluated with Kolmogorov–Smirnov and Shapiro–Wilk tests. Normally distributed variables were given as mean ± standard deviation, and non-normally distributed variables were given as median (25th–75th percentile). Categorical variables were given as frequency (percentage). Differences between groups were determined using an independent sample *t*-test for normally distributed variables and with Mann–Whitney U test for non-normally distributed variables. Relationships between categorical variables were evaluated with Chi-square analysis. *p* < 0.05 was accepted as statistical significance.

## 3. Results

### 3.1. Clinical Results

Statistical analysis of age, the mean week of gestation, the mean birth weight, parity, the first minute APGAR score, the maturity of the newborn, the need for intensive care and the pH values of the umbilical artery blood gas sampling results are presented in Table 1.

### 3.2. Histopathological Results

Placental pathological changes and histopathological findings were evaluated in our study. Placentas of five patients could not be histopathologically evaluated due to lack of proper staining. Deciduitis, perivillous fibrin accumulation, intervillous fibrin, intervillous hemorrhage, infarcts, calcification, laminar necrosis and syncytial node were found to be significantly more common in the study group placentas compared with those in the control group (*p* < 0.001) (Table 2, Figure 1).

### 3.3. Immunohistochemical Results

Immunohistochemical evaluation could not be performed for four of the placentas due to alack of appropriate staining. In the immunohistochemical study performed for the CD56 Ab, 27 (48.2%) of the study group placentas showed low expression and 29 (51.8%) showed high expression. Forty-three (100%) in the control group placentas showed low expression. When compared with the chi-square test, the placentas in the study (PE) group were more stained with CD56 marker, and the difference between the two groups was found to be significant (*p* < 0.001) (Table 3, Figure 2).

In the immunohistochemical study for the ADAM17 Ab, it was expressed as high in all the placentas of the study group patients and low in all the placentas in the control group. ADAM17 Ab stained more in the placentas in the study group, and a significant difference was detected between the two groups (*p* < 0.001) (Table 3, Figure 3).

In the immunohistochemical study performed for the FGF21 Ab, it was expressed as low in 1 (1.8%) placenta in the study group and highly expressed in 55 (98.2%) placentas. FGF21 showed low expression in all placentas in the control group. A statistically significant difference was found between the two groups (*p* < 0.001) (Figure 4, Table 3).

Table 4 presents the comparison of non-severe PE and severe PE placentas in terms of CD56, ADAM17 and FGF21 expression. While 12 (66.7%) placentas of pregnant women with non-severe preeclampsia showed low expression for CD56, 6 (33.3%) showed high expression. While 15 (39.5%) of the placentas with severe PE showed low expression for CD56, 23 (60.5%) showed high expression. When compared to the chi-square test, no significant difference was found between the expression levels of CD56 between the severe PE and non-severe PE groups (*p* = 0.106). All placentas of pregnant women with preeclampsia with and without severe features for ADAM17 showed high expression. For FGF21, 1 (5.6%) of the placentas of pregnant women with preeclampsia that did not show severe features showed low expression, while 17 (97.4%) showed high expression. All of the severe PE group placentas showed high expression for FGF21. A statistical p value could not be determined for ADAM17 and FGF21 expression levels between the groups since the assumptions of the chi-square test could not be met.

## 4. Discussion

Although the pathophysiology of PE is still not clearly explained, it is accepted that abnormal placentation, abnormal spiral artery remodeling, placental insufficiency and endothelial dysfunction are effective in the pathophysiology [3,5,7]. Abnormally released cytokines from the defective placenta, which cause placental insufficiency, also cause widespread systemic endothelial dysfunction [10,11,12,13,14,15,16,17] with various effects of oxidative stress, increased inflammation, genetic and environmental factors contributing to the PE clinic.

CD56 Ab present in intravascular trophoblasts and decidual NK cells [5], which we think will illuminate the pathophysiology of PE, and ADAM17 Ab [8,9,10,11], which has been found to be associated with hypertension and arterial diseases, may be related to PE, which has many studies on its relationship with metabolic syndrome, but there are very limited studies in the literature for its relationship with hypertension. We studied the expression of the FGF21 Ab in pregnant patients with healthy placentas and placentas with PE.

One of the antibodies we studied was CD56. In the study performed by Zhang et al. [19], the placentas of 124 patients with PE and 84 pregnant placentas with complications other than PE (maternal type 2 DM, chorioamnionitis, IUGR, ablatio placenta, etc.) that could cause decidual vasculopathy were compared in terms of CD56 immunostaining. It has been stated that CD56 expression in intravascular trophoblasts can be seen in classical vasculopathy characterized by fibrinoid medial necrosis and acute atherosis. Of 124 preeclamptic placentas, 78 (63%) showed classical decidual vasculopathy (acute atherosis, fibrinoid medial necrosis), and 39 of these placentas were immunostained with CD56. In the immunostaining performed, significant immunoreactivity was observed against CD56 in 37 placentas. The two negative cases were thought to be due to the disappearance of the decidual vessels during fixation. Classical decidual vasculopathy was determined in 84 patients with complications other than PE. Immunostaining was performed with CD56 in the placentas of 28 patients, and immunoreactivity for CD56 staining was observed in all of them. In the present study, CD56 Ab in complicated pregnancy placentas was used to describe CD56-related vasculopathy and mural arterial vasculopathy in spiral arteries. Moreover, CD56 expression was found only in intravascular trophoblasts and decidual NK cells. No other cell types were found to be immunoreactive against CD56 expression in the early implantation site or in the term placenta [19].

In the study of Youssef et al. [20], in which they aimed to investigate the role of NK cells and macrophages in the pathogenesis of PE, 20 healthy third trimester pregnant placentas and 20 third trimester pregnant placentas diagnosed with PE were subjected to CD56 immunostaining, and they found a significant increase in the number of CD56+ decidual NK cells in the PE group. They stated that the functional activity of NK cells mainly depends on the balance of activation and inhibition of KIR receptors by HLA ligands of EVT [20]. In another study by Du et al. [21], they compared CD56+ decidual NK cells isolated from decidual tissues with early and late PE and healthy control groups. It was reported that there were more CD56+ decidual NK cells in flowcytometry in the early and late PE groups compared with the healthy control group, and they were mostly in the early PE group [21]. They stated that the difference might be due to the origin of the biopsy (e.g., decidua basalis versus placental bed). Contrary to these studies, in the study by Lockwood et al. [22], it was reported that a preeclamptic placenta contains fewerCD56 + NK cells [22]. In the study conducted by Williams et al. [23] to compare the leukocyte population density in the placental bed by separating 12 PE, 8 IUGR and 12 control group placentas, it was determined that placentas with PE and IUGR contained less density of CD56+ decidual NK cells, macrophages and lymphocytes [23]. The study of Milosevic-Stevanovic et al. [24] included 30 pregnant women with a diagnosis of PE who gave birth by cesarean section and 20 healthy pregnant women who gave birth by cesarean section. After the placenta was completely removed during cesarean section, the placental bed was scraped with a curette and decidua tissue was removed. CD56 + NK cells, CD68 + macrophages and cytokeratin 7 were analyzed after immunohistochemical labeling of trophoblastic cells. Compared with the control group, the PE group had a significantly lower CD56 + NK cell count and a higher CD68 + macrophage count in the decidua. With an underlying PE in parturients, inadequate implantation occurs early in pregnancy, while clinical manifestations of the disease appear sooner in pregnancy. For this reason, Milosevic-Stevanovic et al. [24] reported that they did not know whether these changes in the composition of immune cells reflect the mechanisms involved in the pathogenesis of preeclampsia or are the result of a disease, since the time between implantation and taking tissue samples for analysis is long. They also commented that not only the number of immune cells, but also their phenotype and cytokine profiles are important [24].

In Zhang’s study [19], CD56 was studied in decidual vessels of pregnant patients who had placental complications and immunoreactivity with CD56 was used to define vasculopathy and mural arterial vasculopathy.

We studied CD56 in placental tissue samples. In our study, in parallel to the studies of Youssef et al. and Du et al. [20,21], CD56 Ab was expressed more in the placentas of patients with PE compared with those in the control group. This suggests that decidual NK cells play an active role in the pathophysiology of PE.

One of the other markers we studied was ADAM17. Ma et al. [9] investigated the mechanism of increased TNF-α production in preeclamptic placentas in their study with immunostaining of five normal and six preeclamptic placentas. They determined whether hypoxia/oxidative stress, an underlying pathophysiology of preeclampsia, would modulate ADAM17 expression and subsequently induce TNF-α production in placental trophoblasts. They found that ADAM17 protein increased in syncytiotrophoblasts of preeclamptic placentas, but there was no change in the expression of its mRNA [9]. In the study of Wang et al. [25], ADAM17 expression was found to be significantly increased in preeclamptic placentas compared with normal placentas. It has been reported that ADAM17 is the major factor that plays a role in TNF-α release and also plays a role in sFlt-1 release [25]. In the study of Yang et al. [26], ADAM10 and ADAM17 were reported to be more expressed in preeclamptic placentas than in normal placentas. In their study, 12 severe PE parturients and 14 control patients were included. They studied the samples with immunohistochemical staining, real-time PCR and Western blot methods [26]. Increased trophoblast TNF-α production is an important component of placental dysfunction in PE [9,25,26]. However, the mechanism of increased TNF-α production in the placentas of pregnant women with PE is still not precisely identified.

In our study, as in the studies of Ma, Wang and Yang, we found that the expression of ADAM17 increased significantly in preeclamptic placentas compared with that in normal placentas. These findings suggest that ADAM17 is involved in the pathophysiology of PE via TNF-α.

ADAM17 has been shown to induce neointimal hyperplasia in the vasculature [27]. After Ang-II infusion, expression of tumor necrosis factor (TNF)-α-converting enzyme (TACE) increases in atherosclerosis and left ventricle, and ADAM17 polymorphism is associated with cardiovascular mortality. Therefore, metalloprotease 17 emerges as a possible therapeutic target for the treatment of hypertension [28]. Ludwig et al. [29] described an ADAM17 inhibitor, GW280264X, which has been shown to block constitutive release of mIL-6R, CX3CL1/fractalkin and chemokine C-X-C ligand 16 [29]. A novel ADAM17 inhibitor has been identified, consisting of SN-4, a zinc-binding dithiol moiety that binds specifically to ADAM17 and inhibits TNF-α release [30]. We also found that ADAM17 expression increased in the placentas of pregnant women with PE and, considering the report by Tateishi et al. [30], it is conceivable that SN-4 could be tested for the treatment of PE in future studies. These agents cause a reduction in inflammation and consequent improvement in hypertension. The use of ADAM17 as a target molecule for drug developments seems to be an effective option for the prevention and treatment of PE.

The last molecule evaluated in our study was FGF21, which was also studied by Stepan et al. [14]. In that study conducted, with 51 healthy and 51 preeclamptic pregnant women, serum FGF21 levels of pregnant women with PE were found to be statistically significantly higher than healthy pregnant women. Moreover, triglycerides, glomerular filtration rate and LDL cholesterol levels have been reported to be independent markers of circulating FGF-21 in pregnant women. In addition, 6 months after birth, serum levels of FGF-21 in PE patients approached the serum levels of control patients [14]. Therefore, they concluded that increased FGF-21 concentrations in PE may be a compensatory mechanism to reduce the adverse vascular and metabolic effects of the disease.

In a study by M Abd Elmagid et al. [31], normal, non-severe PE and severe PE groups, all including 20 parturients, were recruited and serum FGF21 levels were examined. It was determined that the FGF21 level increased by 102.7% in the PEgroup with severe features and by 58.6% compared with the PE group without severe features [31]. In addition, in this study, it was reported that the concentrations of FGF21 in the serum increased significantly up to almost two times compared with the control group, similar to the result we obtained in the expression of FGF21 in placental samples. M Abd Elmagid et al. and Stepan et al. [14,31] stated that the observed increase in PE due to the regulatory effects of FGF21 on intracellular glucose uptake and lipid metabolism may be a compensatory mechanism to reduce the adverse metabolic and vascular effects of PE.

In the study of Dekker Nittert et al. [32], no statistically significant difference was found for placental FGF21 mRNA expression between preeclamptic and normal healthy pregnant placenta. In the same study, serum FGF21 levels were also compared, but no significant difference was found. Moreover, no significant difference was detected for FGF receptors. Dekker Nittert et al. [32] reported that the reason for the lack of increase in FGF21 values in pregnant women with PE may be due to the differences in characteristics such as BMI, insulin resistance and parity of the pregnant women who participated in the study [32].

In a study conducted by RD Semba et al. [13] with 744 patients, they found an independent relationship between FGF21 and hypertension [13]. In this study, 40 non-diabetic obese women exercised five times a week for 3 months and it was determined that there were significant decreases in SBP and DBP, arterial stiffness and serum FGF21 levels. In the same study, it was determined that serum FGF21 levels decreased by 13% after 6 months of angiotensin receptor blocker treatment in 72 patients with end-stage renal disease, and they stated that these findings suggest a possible connection between RAS and FGF21 [13].

The relationship between FGF21 and RAS was reported in former studies [33,34] and the resulting hypertension may occur with similar mechanisms in PE; thus, in the present study we examined placental FGF21 expression, and more intense FGF21 expression was found in placentas with PE. However, in a recently published review, the role of RAS in the pathophysiology of PE originates from AT1-AA, and its role in the pathophysiology of PE has been described as insignificant, with the exception of AT1-AA [35]. Although renin, Ang-II and aldosterone levels are lower in PE than in uncomplicated pregnancies, Wallukat et al. [36] and Siddiqui et al. [37] showed an increased presence of AT1-AA. This mimics the effect of Ang-II in the plasma of preeclamptic women during pregnancy when compared with normotensive pregnant controls [33]. In addition, Ang-II sensitivity of the vascular system increases in preeclamptic pregnancies long before PE becomes clinically evident [33,34,38]. This makes us think that the relationship between FGF21 and RAS is mostly due to AT1-AA.

Former studies in the literature generally studied FGF21 in serum. There is only one study in the literature examining placental FGF21 expression. Finally, in our study, FGF21 expression was found to be significantly higher in the placentas of pregnant women with PE.

Moreover, pathological changes such as preeclamptic placenta-specific deciduitis, perivillous fibrin deposition, placenta giant cell, intervillous fibrin, intervillous hemorrhage, infarct, calcification, laminar necrosis, syncytial node, distal villous hypoplasia and chorionic pseudocyst were determined in the placentas of the PE parturients.

The limitations of our study are that the present study is a single-center study that includes a small number of patients and a control group, whereas the strengths of the present study are that it includes more cases than most of the former studies in the literature, it is prospective and we compared both non-severe PE and severe PE. Considering the low number of studies in the literature regarding the relationship between FGF21, ADAM17 and PE, we think that our study will be a guide for future studies and will be an illuminating study in terms of developing a target molecule for the prevention and treatment of PE.

## 5. Conclusions

In the present study, we found that CD56, ADAM17 and FGF21 expressions were increased in the placentas of pregnant women with PE. The present results are in accordance with the former studies, which reported the roles of CD56, ADAM17 and FGF21 in the development of PE via stimulation of inflammatory processes, macrophage activation, vasodilation defect and increase in cytokines that result in trophoblast invasion. We think that our study results will shed some more light on the pathophysiology of PE. Moreover, it has the potential to have a significant positive effect on maternal and newborn health in pointing out a target for new therapeutic agents.

## Figures and Tables

**Figure 1 medicina-59-01145-f001:**
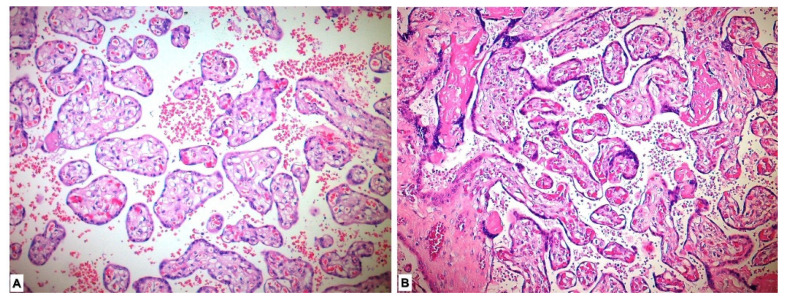
Histopathological view of placental tissues in hematoxylin-eosin sections. (**A**) Normal-appearing chorion villous structures in the placenta in the control group. (**B**) Histopathological changes showing maternal hypoxia such as perivillous-intervillous fibrin deposition and syncytial node in placental tissues of preeclamptic pregnancy.

**Figure 2 medicina-59-01145-f002:**
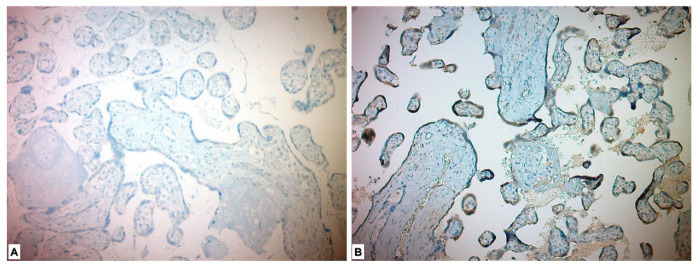
Immunohistochemical expression of CD56 in placental tissues. (**A**) Low CD56 expression in control group placenta. (**B**) High CD56 expression in preeclamptic placenta, more intense in syncytiotrophoblast and cytotrophoblast.

**Figure 3 medicina-59-01145-f003:**
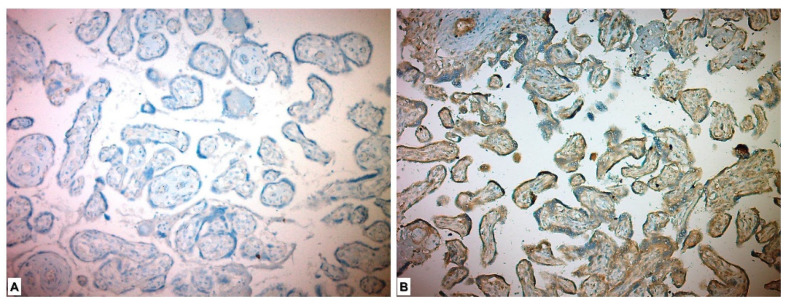
Immunohistochemical ADAM17 expression in placental tissues. (**A**) Low ADAM17 expression in control group placenta. (**B**) High ADAM17 expression in chorionic villous stromal cells, syncytiotrophoblasts and cytotrophoblasts in preeclamptic placenta.

**Figure 4 medicina-59-01145-f004:**
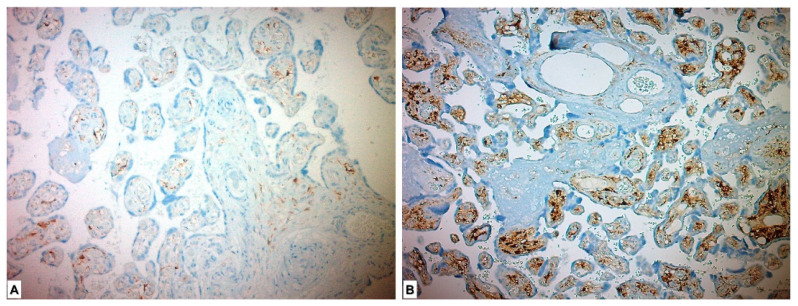
Immunohistochemical FGF21 expression in placental tissues. (**A**) Low FGF21 expression in chorionic villous stromal cells in the control group placenta. (**B**) High FGF21 expression, which is more pronounced in chorionic villous stromal cells of preeclamptic placental tissues.

**Table 1 medicina-59-01145-t001:** Comparison of Study and Control Group Parameters.

Parameters	Study Group (PE) *n* = 60	Control Group *n* = 43	*p*
Age, mean ± SS	29.7 ± 5.7 *	29.9 ± 5.6	0.868 ^§^
Gestational week, median (Q1–Q3)	33 (32–35) **	38 (37–38)	<0.001 ^†^
Birthweight, mean ± SS	2055 ± 667	3122 ± 531	<0.001 ^§^
Parity, *n* (%)			0.474 ^¶^
Nulliparity	32 (53.3)	19 (44.2)	
Multiparity	28 (46.7)	24 (55.8)	
APGAR 1, median (Q1–Q3)	9 (8–9)	9 (8–9)	0.105 ^†^
APGAR 5, ort ± SS	9 ± 0.8	9 ± 0.5	0.025 ^†^
Maturity *n* (%)			<0.001 ^¶^
Premature	50 (83.3)	7 (16.7)	
Term	10 (16.7)	35 (83.3)	
NICU *n* (%)			<0.001 ^¶^
Inpatient	40 (66.6)	8 (19)	
Outpatient	20 (33.3)	34 (81)	
Umb.A. pH, median (Q1–Q3)	7.334 (7.302–7.360)	7.334 (7.309–7.39)	0.417 ^†^

* mean ± SS, S; S: Standard deviation; ** median (Q1–Q3); Q1–Q3: 25–75 percentile; *n*: numbers. ^§^
*t*-test, ^¶^ chi-square test, ^†^ Mann–Whitney U Test, Umb.A.: Umbilical Artery, NICU: Newborn intensive care unit, *p* < 0.05 was considered significant. PE: preeclampsia.

**Table 2 medicina-59-01145-t002:** Placental pathological changes.

Placental Changes	Study Group (PE) *n* = 55	Control Group *n* = 43	*p*
Deciduitis *n* (%)			<0.002 *
Present	10 (18.2)	0	
Absent	45 (81.8)	43 (100)	
Placenta Giant Cell *n* (%)			<0.001 *
Present	7 (12.7)	0	
Absent	48 (87.3)	43 (100)	
Perivillous Fibrin Deposition *n* (%)			<0.001 *
Present	49 (89.1)	0	
Absent	6 (10.9)	43 (100)	
Intervillous Fibrin (%)			<0.001 *
Present	49 (89.1)	0	
Absent	6 (10.9)	43 (100)	
Intervillous Bleeding *n* (%)			<0.001 *
Present	39 (70.9)	0	
Absent	16 (10.9)	43 (100)	
Infarcts *n* (%)			<0.001 *
Present	14 (25.5)	0	
Absent	41 (74.5)	43 (100)	
Calcification *n* (%)			<0.001 *
Present	21 (38.2)	0	
Absent	34 (61.8)	43 (100)	
Laminar Necrosis *n* (%)			<0.001 *
Present	25 (45.5)	0	
Absent	41 (74.5)	43 (100)	
Syncytial Knot *n* (%)			<0.001 *
Present	54 (98.2)	0	
Absent	1 (1.8)	43 (100)	
Distal Villous Hypoplasia (%)			NA **
Present	4 (7.3)	0	
Absent	51 (92.7)	43 (100)	
Chorionic Pseudocyst *n* (%)			NA **
Present	2 (3.6)	0	
Absent	53 (96.4)	43 (100)	

* Chi-square test, NA **: not applicable, *n*: number.

**Table 3 medicina-59-01145-t003:** Comparison of CD56, ADAM17 and FGF21 expressions between study and control groups.

Marker	Study Group (PE) *n* = 56	Control Group *n* = 43	*p*
CD56 *n* (%)			<0.001 *
High Expression	29 (51.8)	0	
Low Expression	27 (48.2)	43 (100)	
ADAM17 *n* (%)			<0.001 *
High Expression	56 (100)	0	
Low Expression	0	43 (100)	
FGF21			<0.001 *
High Expression	55 (98.2)	0	
Low Expression	1 (1.8)	43 (100)	

* Chi-square test, *n*: number.

**Table 4 medicina-59-01145-t004:** Comparison of antibodies between non-severe PE and severe PE groups.

Marker	Non-Severe PE *n* = 18	Severe PE *n*= 38	*p*
CD56 *n* (%)			0.106 *
Low Expression	12 (66.7)	15 (39.5)	
High Expression	6 (33.3)	23 (60.5)	
ADAM17 *n* (%)			NA **
Low Expression	0	0	
High Expression	18 (100)	38 (100)	
FGF21 *n* (%)			NA **
Low Expression	1 (5.6)	0	
High Expression	17 (94.4)	38 (100)	

* Chi-square test, NA **: not applicable, *n*: number.

## Data Availability

The data of the present study can be reached by contacting Rabia Baser Acikgoz via e-mail.

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
