# Peer review of "Investigation of CD56, ADAM17 and FGF21 Expressions in the Placentas of Preeclampsia Cases"

_medicina, 2023, doi:10.3390/medicina59061145_

Round 1

Reviewer 1 Report

Thank you very much for the opportunity to review the manuscript entitled “Investigation of CD56, ADAM17 and FGF21 Expressions in Placenta in Preeclampsia Cases

Abstract

L15 – for antibodies, please use the acronym “Ab” and modify each appearance throughout the abstract and, afterwards, throughout the manuscript

L16 – please consider to reformulate “preeclamptic and healthy pregnant placentas” with pregnant patients with healthy placentas or placentas with PE” and, afterwards, throughout the manuscript.

L20 – please clarify the institution where the patients were admitted – it is either a University or the Faculty of that University?

L19-25 – please reformulate the Material and Methods section, in order to be more clearer what are the inclusion and exclusion criteria

L25-26 – please reformulate, there is not clear

L29-30 – Please reformulate, because there seem to be an observational study, while you are stating here that “…expression increased…” and you not mention the cause of the increase.

L30-34 – please reevaluate if this information are worthy to be part of the abstract or if they are better to be moved to the Discussion or Conclusion sections.

            Introduction

L42 – please rephrase because the meaning of the “interval of 4 hours” is missing

L43 – if you use an acronym once in the manuscript, please check the whole manuscript

L76 – there is a “i” that should be corrected, because it has no meaning

Material and Methods

L114-116 – please mention the number of the Ethical Committee approval/

L116-118 – if you use an acronym once in the manuscript, please check the whole manuscript

L116-120 – please explain why you excluded the patients with pre-existing placental pathologies

L131 – please reformulate “enfarct”

Results

L173-213 – please make more uniform the data reported. Eg – the number of patients as numbers. Moreover, you reported the mentioned data as Table 1, so this information are redundant also as text, please reformulate and report only once the data.

L214-227 and L283 – pleare redo the tables, because their formatting is lacking.

L228-229 – please mention the Figure in the text and move it close to its mentioning.

L263-264 – please move the Table close to its mentioning in the text.

Discussions - please reformulate  this section and take into consideration the following recommendation

L285-296 – please add references regarding the reported data

L298-315, L316-321, L322-327, L328-333, L334-348, L355-361, L362-365, L366-369, L390-398, L399-408, L409-415, L416-423, L424-436 – please report data from other study in comparison with the data reported by your research results

Conclusions - please reformulate  this section and take into consideration the following recommendation

L452-488 – please reevaluate this section because it has elements that better fit in the Discussion section.

References - Some references are old (before year 2000), please reconsider them. Please redo the references to the Journal recommendations.

The level of English should be improved.

Author Response

Dear Reviewer,

Thank you for your kind interest and time for our manuscript. Your valuable recommandations are revised on the manuscript. Please find atteched the revisions in the below attachment file.

Best Regards,

Dr. Rabia Baser Acikgoz

Reviewer 2 Report

The manuscripts are well written but the authors need to edit the English in their manuscript, there are areas that need their attention. In addition Table 2 needs reformatting, the contents are scattered everywhere. There is also need for formatting the lines, for example lines from 278-283 are spaced differently from the rest of the lines in the document. The major problem with this good study is the small sample size, hence some non normally distributed data. The use of non parametric tests for the data has reduced the quality of the results and the interpretation thereof. However, it is a good paper that should be published upon addressing the minor issues presented. 

The quality is good, but the paper needs some proofreading and reformatting the lines in some areas.

Author Response

Dear Reviewer,

Thank you for your kind interest and time for our manuscript. Your valuable recommandations are revised on the manuscript. Please find attached  below the revisions made on the manuscript.

Best Regards,

Dr. Rabia Baser Acikgoz

Round 2

Reviewer 1 Report

Thank you very much for revising the reccomended aspects. The refference cand be improved, due to the high number of older than years old used.

Minor improvements can be made, but the current form is satisfacatory.